**communications** sustainability

# Measuring carbon capability beyond the carbon footprint
Alisa Ghura[1], Sam Hampton [1,2] ✉ & Lorraine Whitmarsh[2]

Meeting climate action targets requires both individual and systemic change. Behaviour change can contribute to system change through actions in the public sphere, including influence and citizenship. However, current measurement approaches, such as personal carbon footprints, emphasise individual consumption and underrepresent public-sphere contributions. This study operationalises a framework which integrates individuals' motivation and capacity to reduce emissions within broader systems of provision. We present a methodology to quantify public-sphere actions and capabilities alongside consumption behaviours, generating a comprehensive capability score. Applying this approach to a representative survey UK residents (N = 2001), we find moderate-to-low climate action capability, with the lowest scores in transport, food, and civic domains. Regression analyses indicate gender, education, and climate knowledge predict higher capability. This methodology offers an integrated tool to assess both private and public climate actions, informing strategies for more effective engagement and policy interventions.

Tackling climate change and reducing global greenhouse gas (GHG) emissions requires large-scale behavioural and structural change[1]. However, instead of recognising the need for both individual and systemic change in promoting low-carbon behaviour, these are often framed in opposition. One account focuses on individuals' shortcomings ('i-frame') and solutions involving widespread behavioural change. Another attributes climate change to the failure of systems ('s-frame')[2].

This individual vs. systemic framing has a long history in social sciences, and persists in climate discourse[3]. For instance, the personal carbon footprint (CF) has been critiqued for deflecting attention from the irresponsible activities of extractivist industries and the need for deep-rooted reform of economic and political systems, which allow for unabated use of fossil fuels[4]. Nonetheless, globally, populations express high concern about climate change and a desire to take individual action[5]. Unfortunately, individuals tend to misunderstand which of their behaviours contribute most to climate change. On average, people *underestimate* the most impactful climate actions, such as eating less red meat, and *overestimate* activities such as recycling or switching off lights[6].

CF tools can help to build climate literacy and promote behaviour change. But there is little evidence of widespread use or impact[7], and they are criticised on methodological and theoretical grounds[8]. CF calculators poorly account for how systems of provision shape GHG emissions of various lifestyles[9]. CFs also focus on everyday consumer choices and habits (e.g. turning off lights) rather than high-impact, one-off behaviours (e.g. replacing boilers), and neglect the multifaceted ways in which people can and do engage in climate action, including in the public sphere (e.g. voting or

influencing behaviours[10]). Some studies have sought to address these weaknesses by analysing relationships between carbon literacy and footprints[11], and some attempts have been made to quantify the impacts of investments and political choices[12]. One study which quantified systemic factors found that a *homeless* person's CF in the US exceeded the global average[13].

This article builds on recent work highlighting the multiple capacities of individuals to drive climate action in the private (consumption) and public (social, political) spheres by adopting the 'carbon capability' (CC) framework. CC is defined as an individual's ability and desire to reduce their GHG emissions within broader systems of provision[14]. Capabilities are determined by: (1) decision-making (e.g. motivations, knowledge, skills), (2) behaviours and practices and (3) systems of provision. The framework has been used to evaluate the capabilities of the UK public[15,16], urban citizens[17] and farmers[18] in Eastern China, and wealthy British people[19]. CC studies have taken different approaches to conceptually and quantitatively evaluating individuals' ability to reduce their carbon emissions[15,16,18,20,21]. Wei and colleagues' model depicts *stages* of CC, progressing from foundational personal values, to engagement and consistent behaviour over time, and finally social *influence*[20]. This *linear* conceptualisation is both a strength and a weakness. It highlights how CC often involves a process or journey experienced by individuals, but it fails to capture situations where low-carbon behaviours are enacted *without* the need for underpinning values relating to environmental sustainability. Motivations for active travel, or vegetarianism, need not be for climate reasons, for example.

[1]Environmental Change Institute, University of Oxford, Oxford, UK. [2]Department of Psychology, University of Bath, Bath, UK. ✉e-mail: sah53@bath.ac.uk

Seeking to develop and validate a scale, Horry and colleagues focus on knowledge, behaviours, and attitudes, incorporating items related to observed climatic changes, personal practices, and support for proactive government intervention[15]. In aiming for a concise set of indicators, this approach necessarily entails selectivity, and as a result, certain dimensions of CC may be underrepresented. This underscores an inherent trade-off in scale development between parsimony and comprehensive coverage of the diverse elements comprising CC.

Studies of CC have also been critiqued for bias towards methodological individualism, due in part to the widespread use of self-report survey methods[16]. Rather than focus on the responsibility of individuals to 'engage with systems of provision'[14], there is a need for CC research which examines how systems of provision constrain and enable low-carbon choices[9], and methods which capture community-scale activity for building collective CC (which can include survey research as well as other methods)[16]. Nonetheless, individual-level survey methodologies provide unique insight into how people experience and interpret structural and systemic constraints, allowing analysis of the intersection between personal agency and broader social and institutional contexts[22].

This study defines CC as a synthesis of an individual's demonstrated ability and expressed commitment to reduce emissions, measured through both quantifiable CFs and multi-domain behavioural, attitudinal, and participatory indicators, all situated within the constraints and opportunities presented by broader systems of provision. It employs Hampton and Whitmarsh's six *domains* for CC[16,23]. Four involve consumption-based activities with direct GHG emissions (household energy, transportation, food, other 'shopping'), and two incorporate indirect public-sphere behaviours and attitudes: citizenship, and influence[16,23]. All six domains are influenced by interdependent individual, political, social, and structural factors. For example, knowledge of how to take low-carbon actions is important, but insufficient if individuals are unable to access the goods and services which can help them reduce their environmental impacts (e.g. energy-efficient appliances, healthy low-carbon food options, safe cycling and walking infrastructure), or feel constrained by social norms and expectations. These individual and system-level interactions are at the core of CC[14].

Despite widespread use of empirical data to evaluate CC[14,15,20], no studies have incorporated quantitative assessments of actual consumption-based footprints, nor is there an accepted method for incorporating public-sphere behaviours (e.g. citizenship) into CC assessments. This article addresses this gap by operationalising the CC framework to create a comprehensive and statistically rigorous approach to quantifying CC. There are strong conceptual and practical arguments for developing a scoring methodology. A numerical or categorical score enables systematic comparison across individuals, demographic groups, and populations, making disparities in capability visible and thereby informing more targeted and equitable policy interventions[24]. It also allows for the identification of strengths and weaknesses across domains, guiding the effective allocation of resources and the design of educational and behavioural initiatives. Quantified metrics facilitate the tracking of CC over time, providing a means to evaluate progress and the impact of interventions. Furthermore, scoring offers meaningful feedback to individuals and communities—similar to CF calculators—supporting self-reflection, encouraging behavioural shifts, but also highlighting where systematic barriers impede progress. Whereas CFs focus on individual consumption behaviours, a CC score may prompt greater engagement in public-sphere actions, such as community organising or political advocacy[23].

Using data from a nationally representative survey from the UK ($N = 2001$), the article addresses three research questions:

1. How can carbon capability be calculated into a comprehensive score?
2. How carbon capable are UK residents, and how does capability vary across the six domains?
3. What factors predict carbon capability?

Moving beyond individual-vs-systemic framings, this article introduces a novel methodology for generating a CC Score comprised of two components: (1) CF (measured in $tCO_2e/yr$) and (2) 'capability scores' (measured using a letter grade A–F). Our CF methodology emulates and improves upon a widely used calculator[25,26] and results are validated against government estimates of UK household footprints. Capability scores are organised using the six CC domains, reflecting the main categories of consumption-based emissions as well as two key domains of non-consumption behaviours[16]. Likert-scale items span the four components outlined above, for breadth and consistency across the domain.

## Results

This section addresses the RQs by: (1) demonstrating and validating the methodology outlined above on our sample of UK residents, (2) assessing the CC of this representative sample, and (3) identifying key predictors of CC.

Assessing and validating CC of UK residents: capability scores for each domain are measured using 12 items (see 'Methods'), with three items chosen to assess crosscutting 'components' of CC:

- Individual traits (e.g. attitudes, knowledge, skills)
- Choices and behaviour
- Broader engagement with carbon governance and social norms
- Structural capacity (ability to act within systems of provision).

Across all six domains, $\bar{x} = 37.75$, which equates to C on the A–F scale (Table 1; Fig. 1). This aligns with recent research findings of low-to-mid-level CC in the UK[16].

Capability scores are broken down by domain, on a scale of $-36$ to $36$ (Fig. 2). Average scores are highest in household energy and lowest in citizenship.

The 12 items used to measure domain-specific capability scores are found to be reliable ($\alpha > 0.7$ for each), which indicates consistency between each of the four components. However, comparing mean scores for the components (combining domains) reveals key differences (Fig. 3). Highest aggregate scores are for *individual traits* (attitudes, knowledge, skills, etc.) and *broader engagement*. The *behaviour* & *choices* component (most directly related to consumption emissions) scores lowest.

Analysing component scores by domain reveals key differences (Fig. 4). Respondents report pro-environmental *behaviours and choices* with respect to three consumption domains, but these scores are negative for transportation, citizenship and influence. *Structural capacity*, which refers to barriers, infrastructures, perceived behavioural control and access to support systems, also varies by domain, and the mean score for this component is negative in the *influence* domain. Respondents perceive greater barriers to influencing others on climate change, compared with acting themselves.

### Assessing and validating the carbon capability of UK residents: Carbon Footprints

Mean CF values are 10.06 $tCO_2e$/year (Table 2). This is consistent with other studies and higher than UK government figures due to our more comprehensive scope (see 'Methods'). Figure 5 is a histogram showing the distribution of respondents' CFs with a leftward skew. The mean is higher than the median due to emissions inequality, as observed elsewhere[27,28].

**Table 1 | Capability score summary statistics**

| Minimum | 1st quartile | Median | Mean | 3rd Quartile | Maximum | Standard deviation |
|---|---|---|---|---|---|---|
| −118.00 | 7.00 | 39.00 | 37.75 | 71.00 | 183.00 | 49.72 |

**Fig. 1 | Distribution of capability scores.** Scores are assigned letter categories.

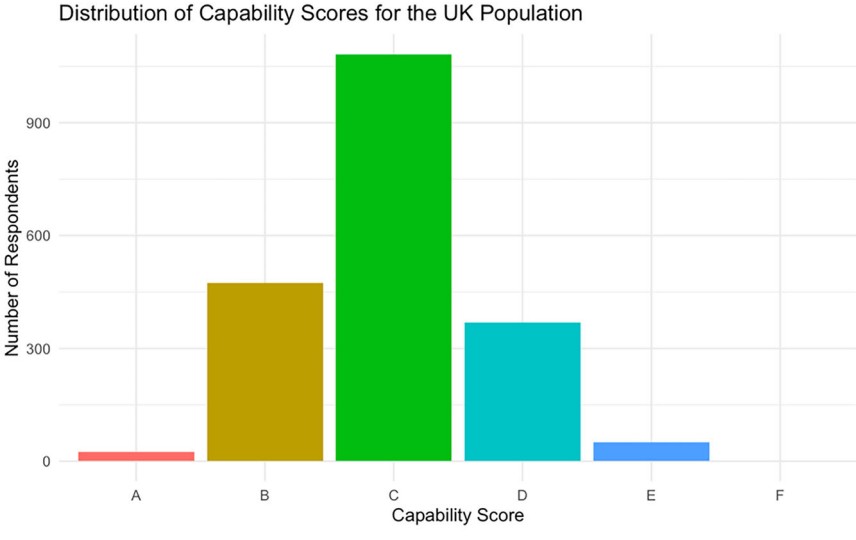

Distribution of Capability Scores for the UK Population

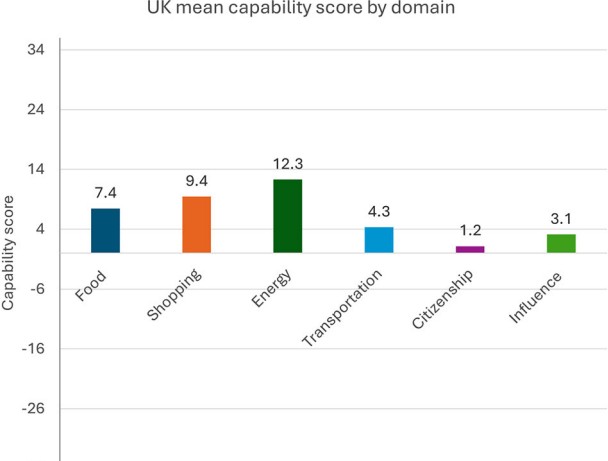

**Fig. 2 | Capability score by domain.** Average capability scores across the six domains of carbon capability.

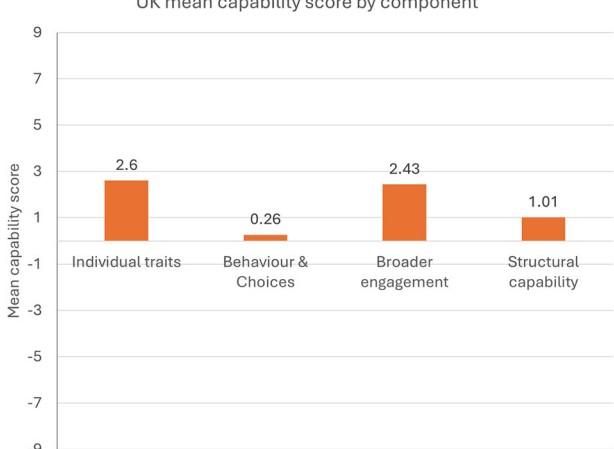

**Fig. 3 | Mean capability component scores.** Four components cut across domains.

Transportation makes up almost half of the average CF (4.9 $tCO_2e$ see Fig. 6). This is higher than the overall share of transport towards UK emissions[29], reflecting the major contribution of individual travel behaviours. Mean household energy CF is 1.1 $tCO_2e$ (11%). This is slightly lower than Office for National Statistics (ONS) estimates of 1.4 $tCO_2e$[30], but expected based on the training data used for the novel modelling approach[31].

Overall, the capability score and CF results are consistent with the literature, indicating that the methodology is a reliable tool for quantifying CC.

Findings also reveal a significant relationship between capability scores and CF calculations when controlling for demographic variables and climate knowledge. Table 3 shows the regression output where the dependent variable is CF, and in Table 4 the capability score is the dependent variable. A one-point increase in capability score is associated with a 0.01 $tCO_2e$ decrease in CF (Table 3). A 1 $tCO_2e$ increase in CF is associated with a decrease in capability score by 0.60 points (Table 4). This aligns with literature indicating an inverse relationship between CC and CFs[14,15,18,20].

### Predictors of carbon capability

Addressing RQ3, this section identifies predictors of capability scores and CFs independently, using a set of demographic variables and a climate knowledge variable based on four questions (see ref. 32, Qs 13–16). Climate knowledge and level of education are distinct variables and found to be weakly correlated ($R = 0.12$) and not collinear.

### Regression: capability score

Regression analysis finds that gender, education and climate knowledge are significant predictors of capability score (Table 5). Climate knowledge has the largest standardised coefficient (0.18), followed by education, where each level of attainment (e.g. GCSE or lower; A-levels, etc.) is associated with an 8.92-point increase in capability score. Being female is associated with a 5.75-point increase.

### Regression: carbon footprint

Gender, age, level of education, income, household ownership, working, and climate knowledge are each significant predictors of CFs (Table 6). Income is the strongest standardised predictor, and CFs increase by 0.7 $tCO_2e$ with each income decile. Being female is associated with a decrease in CF by 0.69 $tCO_2e$, and age is weakly correlated with CF. Individual footprints increase with level of education, and being a homeowner is associated with an increase in CF of 0.99 $tCO_2e$. When controlling for other variables, results show that individuals who work either part-time or full-time have larger CFs. Our four-item test of climate knowledge also significantly predicts CF, indicating that greater knowledge is associated with lower consumption emissions.

**Fig. 4 | Capability scores by domain and component.** Combines component and domain scores.

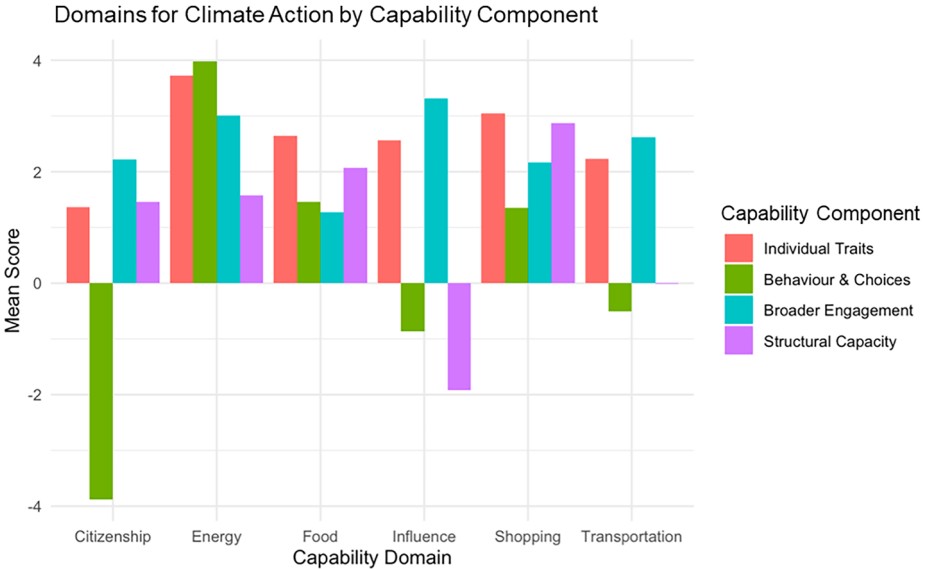

**Table 2 | Carbon footprint summary statistics**

| Minimum | 1st quartile | Median | Mean | 3rd quartile | Maximum | Standard deviation |
|---|---|---|---|---|---|---|
| 2.42 | 6.12 | 8.30 | 10.06 | 12.11 | 49.39 | 5.96 |

**Fig. 5 | Distribution of carbon footprints based on survey analysis.**

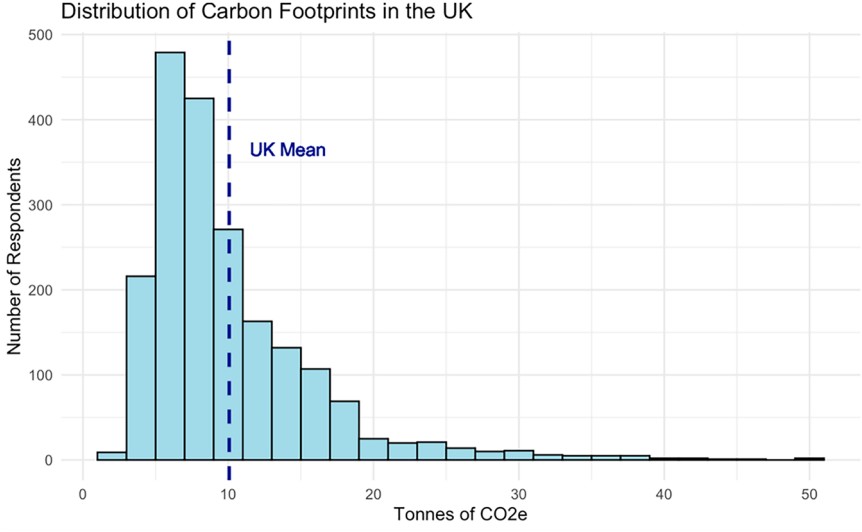

Figure 7 compares standardised coefficients from both regression models. Higher CC is indicated by a combination of positive capability score and negative CF coefficients. Being female and performing well in the climate knowledge test both significantly predict higher levels of CC.

## Discussion

This study develops and demonstrates a novel methodology for quantifying CC. Responding to the need to move beyond the i/s-frame dichotomy in debates about climate action, it incorporates CF assessments, measures of individual traits, behaviours and choices, engagement with systems of provision, and structural factors.

The methodology produces two measures: an individualised CF expressed in tonnes of GHG emissions per year, and a capability score

ranging from A to F. Applying this methodology to a representative sample of UK residents ($N = 2001$), we find a mean CF of 10.06 tCO₂e, and a mean capability score of 'C'. This aligns with other estimates of individual CFs, including government statistics and previous CC research, which finds limited levels of capability in the UK. Statistical analyses show that CF and capability score have an inverse relationship, and each is significantly predicted by several demographic variables as well as a measure of climate knowledge. Unlike carbon literacy assessments or enhanced calculators focused on individual knowledge or pro-environmental behaviours alone, our capability score combines physical, cognitive, and systemic components to capture a fuller picture of climate action capacity.

The mean CF (10.06 tCO₂e) is more than twice the global average CF, highlighting significant inequity in household emissions[19,33]. Within the UK,

income and home ownership are positively correlated with CF, and not significantly correlated with capability scores. Whereas Moorcroft and colleagues[19] point to wealthy people's knowledge of climate, support for policies, and ability to adopt low-carbon technologies as contributing

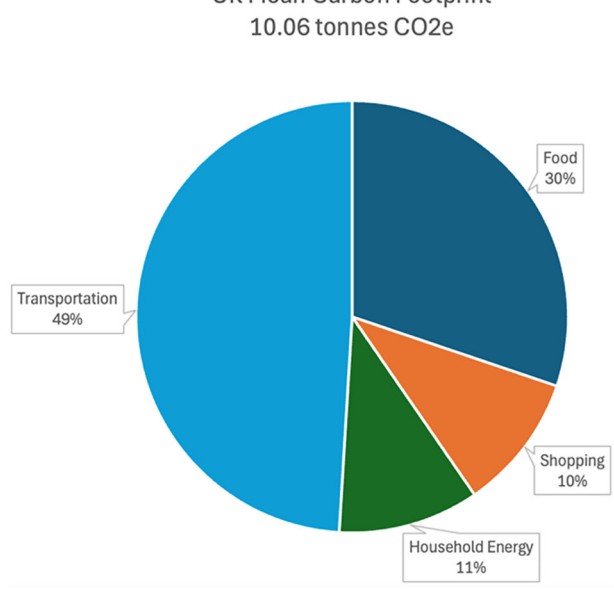

**Fig. 6 |** Mean carbon footprint by consumption domain.

positively towards their CC, we find their outsized CFs are *not* offset by positive capability scores. High income is the strongest predictor of poor capability, giving support to arguments for policy to urgently readdress emissions inequality[10,19].

Climate knowledge is significantly associated with higher capability scores and lower CFs, even when controlling for educational attainment. This finding gives support to initiatives designed to boost climate knowledge, such as the Carbon Literacy Project[34].

Analysis of capability scores finds an average C score (on a scale from A to F), and component scores indicate that *individual traits* (e.g. attitudes, knowledge, and skills) and *broader engagement* with systems of provision contribute most positively, while scores for *behaviour and choices*, and *structural factors* are lower on average. Notwithstanding self-reporting bias, which may lead respondents to rate their own attitudes, knowledge, and social and political engagement favourably, these findings indicate the need to reorientate systems of provision to make low-carbon behaviours and choices easier and more affordable. The UK public says they are open to pro-environmental behaviour change but face structural barriers to doing so.

Breaking down CC by domain highlights priority areas for policy. Transportation contributes the largest share of the average household CF (49%) and has the lowest mean capability score of the four consumption domains. Component scores for transport (Fig. 4) reveal structural factors—particularly inadequate infrastructure and limited alternatives—as the binding constraint on low-carbon travel, rather than a lack of attitudes or knowledge.

Surprisingly, the best performing domain is household energy (B" score, mean = 12.3), which contrasts what is known about the poor energy performance of the UK housing stock and the multifaceted challenges

**Table 3 | Regression output where the carbon footprint is the dependent variable, and the capability score is included as an independent variable**

| | Estimate (B) | Standardised (beta) | Robust std. error | t value | p value |
|---|---|---|---|---|---|
| (Intercept)*** | 5.27 | NA | 0.67 | 7.88 | 0.00 |
| Capability score** | −0.01 | −0.06 | 0.00 | −2.90 | 0.00 |
| Female* | −0.64 | −0.05 | 0.25 | −2.55 | 0.01 |
| Age* | 0.02 | 0.05 | 0.01 | 1.99 | 0.05 |
| Level of education*** | 0.64 | 0.09 | 0.18 | 3.54 | 0.00 |
| Income decile*** | 0.70 | 0.30 | 0.06 | 11.59 | 0.00 |
| Household ownership*** | 0.97 | 0.08 | 0.29 | 3.36 | 0.00 |
| Working*** | 1.19 | 0.10 | 0.28 | 4.30 | 0.00 |
| Climate knowledge. | −0.22 | −0.04 | 0.11 | −1.99 | 0.05 |

Model fit: $F_{(8,1992)} = 51.09$, $p = 0.00$, $R^2 = 0.17$, Adj. $R^2 = 0.17$. $*p < .05$, $**p < .01$, $***p < 0.001$.

**Table 4 | Regression output where the capability score is the dependent variable, and carbon footprint is included as an independent variable**

| | Estimate (B) | Standardised (beta) | Robust std. error | t value | p value |
|---|---|---|---|---|---|
| (Intercept) | 4.92 | NA | 5.90 | 0.83 | 0.40 |
| Carbon Footprint** | −0.60 | −0.07 | 0.21 | −2.85 | 0.00 |
| Female* | 5.34 | 0.05 | 2.25 | 2.37 | 0.02 |
| Age | 0.02 | 0.01 | 0.08 | 0.33 | 0.74 |
| Level of education*** | 9.26 | 0.15 | 1.56 | 5.95 | 0.00 |
| Income decile | −0.10 | −0.01 | 0.50 | −0.21 | 0.83 |
| Household ownership | −2.01 | −0.02 | 2.49 | −0.81 | 0.42 |
| Working | 1.84 | 0.02 | 2.44 | 0.76 | 0.45 |
| Climate knowledge*** | 8.14 | 0.18 | 0.99 | 8.25 | 0.00 |

Model fit: $F_{(8,1992)} = 17.50$, $p = 0.00$, $R^2 = 0.07$, Adj. $R^2 = 0.06$. $*p < .05$, $**p > .01$, $***p < .001$.

**Table 5 | Regression output: capability score as dependent variable**

|  | Estimate (*B*) | Standardised (beta) | Robust std. error | *t* value | *p* value |
|---|---|---|---|---|---|
| (Intercept) | 1.77 | NA | 5.82 | 0.30 | 0.76 |
| Female* | 5.75 | 0.06 | 2.25 | 2.55 | 0.01 |
| Age | 0.01 | 0.00 | 0.08 | 0.18 | 0.86 |
| Level of education*** | 8.92 | 0.14 | 1.56 | 5.73 | 0.00 |
| Income decile | −0.53 | −0.03 | 0.48 | −1.09 | 0.27 |
| Household ownership | −2.60 | −0.03 | 2.49 | −1.05 | 0.30 |
| Working | 1.13 | 0.01 | 2.42 | 0.47 | 0.64 |
| Climate knowledge*** | 8.31 | 0.18 | 0.99 | 8.42 | 0.00 |

Model fit: $F_{(7,1993)} = 18.62$, $p = 0.00$, $R^2 = 0.06$, Adj. $R^2 = 0.06$. *$p < .05$, ***$p < 0.001$.

**Table 6 | Regression output—carbon footprint as dependent variable**

|  | Estimate (*B*) | Standardised (beta) | Robust std. error | *t* value | *p* value |
|---|---|---|---|---|---|
| (Intercept)*** | 5.26 | NA | 0.67 | 7.86 | 0.00 |
| Female** | −0.69 | −0.06 | 0.25 | −2.73 | 0.01 |
| Age* | 0.02 | 0.05 | 0.01 | 1.98 | 0.05 |
| Level of education*** | 0.57 | 0.08 | 0.18 | 3.16 | 0.00 |
| Income decile*** | 0.70 | 0.30 | 0.06 | 11.63 | 0.00 |
| Household ownership*** | 0.99 | 0.08 | 0.29 | 3.43 | 0.00 |
| Working*** | 1.18 | 0.10 | 0.28 | 4.27 | 0.00 |
| Climate knowledge* | −0.29 | −0.05 | 0.11 | −2.65 | 0.01 |

Model fit: $F_{(7,1993)} = 56.85$, $p = 0.00$, $R^2 = 0.17$, Adj. $R^2 = 0.16$. *$p < .05$, **$p < .01$, ***$p > .001$.

relating to retrofit[35,36]. This discrepancy likely reflects the survey items (Q24), which emphasised everyday practices—such as switching off appliances and moderating heating—rather than more demanding measures like heat-pump installation or deep retrofit. In other words, capability appears relatively high for low-cost actions, while structural and financial barriers remain for large-scale improvements. Future iterations of the methodology could therefore 'raise the bar' by incorporating these higher-impact items, providing a more stringent test of capability in this domain.

In the food domain, respondents report relatively high levels of skill and awareness about low-carbon diets, but food still makes up 30% of an individual's emissions. This suggests the main constraint is access and affordability of low-carbon options, not a lack of knowledge. Scores are lowest for broader engagement with carbon governance and social norms. Given how diet choices are culturally embedded and influenced by accessibility, there is a need for measures to encourage the consumption of low-carbon foods[37]. 'Midstream' interventions can be as important as 'upstream' policy measures for achieving this[38]. These include making low-carbon foods widely available in supermarkets and restaurants, product labelling, and using choice architecture to influence behaviours. Responsibility for midstream interventions does not just fall to central government: businesses, community organisations, and local governments must be mobilised to build low-carbon dietary norms.

Lastly, respondents have low capability in influence and citizenship. These domains are not associated with household CFs but are crucial for delivering the systemic change needed to meet the UK's climate goals. The *behaviour and choices* component scores are notably low, indicating limited uptake of concrete civic and financial actions (e.g. talking about climate change, investing in green pensions, participating in environmental projects or activism). Encouraging these behaviours must involve a range of civil society organisations and businesses, as well as governments. By contrast, broader engagement scores are relatively high, suggesting general support for government action both domestically (making low-carbon choices easier and more attractive), as well as internationally, adopting a leadership position in global climate negotiations. This points to an 'action gap': respondents largely endorse climate action in principle but do not routinely translate this into personal civic behaviour.

The novel methodology developed and tested in this article has several limitations. Like nearly all CC studies, it relies on survey data, with associated limitations[39]. Quality control procedures identified and removed nonsensical and rushed responses, but these cannot fully ensure that all responses are honest and accurate. There are also limitations to self-reporting, including where individuals may not have full knowledge or access to relevant information, or want to present themselves favourably. All CF methodologies must balance trade-offs between accuracy and assumptions, bearing in mind that respondents have limited time and information. This study adheres to Birnik's[40] principles and improves upon the popular WWF calculator in several ways. It mitigates the known unreliability of self-reported estimates in energy, food, and 'shopping' domains[41] by using secondary data sources and modelling.

Survey design aimed to minimise bias by using straightforward questioning with consistent wording across domains and standardised Likert scales. Item selection was informed by an extensive review of CC and environmental psychology literature, but nonetheless involves a degree of subjectivity. For capability score calculations, items were assigned equal weight to avoid normative judgement. However, this approach is not infallible: it could be argued that an individual's response to a question about flying behaviours says more about their CC than a measure of social capital, for example. But the CC framework deliberately goes beyond readily quantifiable measures of consumption emissions, and our methodology builds on recent efforts to highlight the importance of individuals' multiple roles and public-sphere behaviours in accelerating widespread climate action[3,14]. In summary, the CC methodology—like CF calculators—is a model involving reasoned, evidence-based assumptions.

Recent advancements in CF methodologies have incorporated elements such as carbon literacy, political agency, and green investment decisions[11,12]; nevertheless, these innovations remain methodologically segmented, often privileging discrete cognitive, behavioural, or public-sphere domains without fully integrating the composite nature of individual climate capacity. The methodology developed here builds on these efforts by operationalising CC as a multidimensional construct, combining CFs with measures of motivation and knowledge, the adoption of influence and citizenship behaviours, and structural enablers/constraints. It contributes a novel evaluative model that reflects the multiple roles individuals play in climate action—beyond consumption—and provides decision-makers and communities with robust comparative data.

**Fig. 7 | Capability score and carbon footprints.**
Comparison of standardised coefficients, showing significant predictors across regression models.

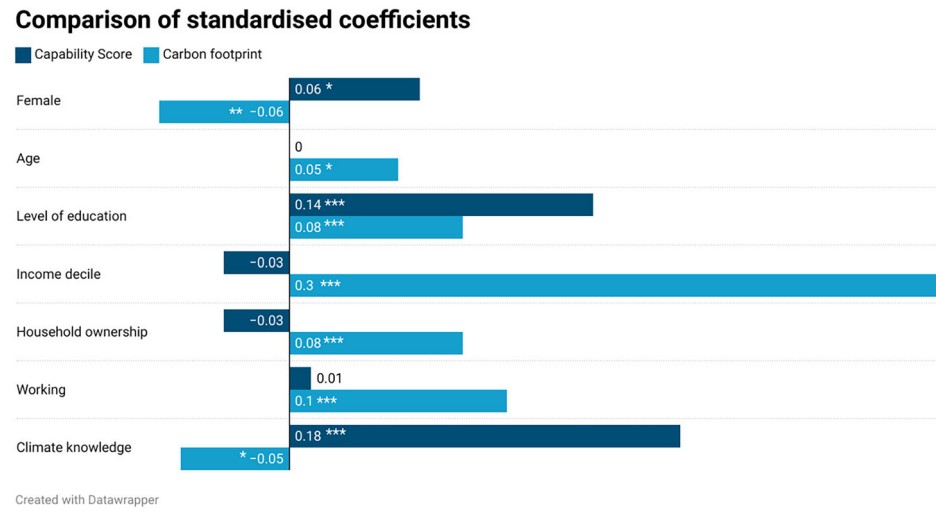

### Comparison of standardised coefficients

■ Capability Score ■ Carbon footprint

Taken together, these findings have clear implications for policy and practice. Overcoming structural barriers to transport capabilities requires investment in public transport and active travel infrastructure, while there is a need to normalise low-carbon diets, including through midstream interventions. The strong association between income and CFs also highlights the need to address emissions inequality, ensuring that the highest-emitting households bear proportionate responsibility for mitigation. Beyond consumption choices, the poor performance in influence and citizenship suggests that civic, institutional, and financial channels of climate engagement need strengthening if climate targets are to be achieved.

Conceptually, this study advances the literature on CC by operationalising it as a multidimensional construct that integrates measurable CFs with behavioural, cognitive, structural, and public-sphere components. This goes beyond prior focus on either carbon literacy scales or household calculators, offering an empirically grounded tool for capturing the interplay between individual agency and systems of provision. In doing so, it provides a means of empirically bridging i-frame and s-frame debates[2], reframing how the roles of individuals in systemic transitions are assessed.

Future research should further refine the measurement of capability across domains, for example, incorporating more ambitious indicators of high-impact actions such as heat-pump adoption or sustained civic participation. Applying the framework across countries, social groups, and community contexts would provide comparative insights into how capability is differently enabled or constrained. Longitudinal and mixed-methods designs could shed light on how capability evolves in response to changing policies, infrastructures, and cultural norms. Advancing this agenda will help to more precisely understand the conditions under which individuals and communities can contribute to accelerating systemic decarbonisation.

## Methods
### Research questions and validating methodology
Responding to *RQ1*, the goal was to create a methodology which accurately estimates individuals' CFs and their ability to reduce emissions, validated against existing estimates and methodologies. By all accounts, household emissions have been steadily declining in the UK since 1990[29], however, estimates of average annual emissions vary significantly. One approach is to divide the UK's overall GHG emissions by the UK population, producing estimates of per capita footprints between 10 and 13 $tCO_2e$[42]. However, even this approach yields different conclusions. The Climate Change Committee attributes around 40% of UK emissions to householders[29], while the ONS puts this figure at 27%[43]. ONS calculations produce an average of 7.3 $tCO_2e$ per household, as they exclude emissions related to exports, aviation, and shipping. Earlier UK studies found notably higher household emissions.

Using an input-output model, Druckman and Jackson found $\bar{x} = 26$ $tCO_2e$ using 2004 data[44], and Buchs and Schnepf calculated $\bar{x} = 20.18$ $tCO_2e$ for the period 2006–2009[45].

In terms of the capability score methodology, there are fewer datasets from which to validate it. However, two representative studies have measured elements of CC in the UK. Using a 2008 survey, Whitmarsh et al.[14] found that CC was relatively limited and that environmental impact was not a major consideration in the daily lives of most. Hampton and Whitmarsh[16] concluded from their 2022 survey that CC was improving slowly but remained insufficient for achieving national climate goals. We therefore expected to find low-to-middle range CC in our study. Lastly, the logic of the CC framework suggested that we might find an inverse relationship between the capability score and CFs. Given that the CC framework (see four major components above) included choices and behaviour, it was reasonable to assume that people with high capability scores would have lower CFs, on average. The multivariate statistical methods used to test these assumptions are outlined below.

### Designing the survey instrument and calculating carbon capability
The goal of the survey was to assess the CC of respondents and factors which influence CC. It was structured around the six domains of climate action (household energy, transportation, food, shopping, influence, and citizenship)[16] and the four components of capability (attitudes, knowledge, skills; choices and behaviour; broader engagement with carbon governance and social norms; and structural capacity). See ref. 32 for the full survey.

### Capability score
To enable comparison across domains, each was assessed using a standardised matrix of 12 items, evenly divided into the four components of CC (see Table 7 for transportation as an example). *Individual traits* captured attitudes, values, skills, and self-assessed knowledge relevant to the domain (e.g. perceived importance of reducing environmental impacts, confidence in using low-carbon alternatives). *Choices and behaviour* comprised self-reports of specific practices with direct emission consequences (e.g. frequency of car use for short trips, adoption of household energy-saving measures, avoidance of food waste). *Structural capacity* items assessed access to infrastructure, affordability of alternatives, and contextual constraints such as caring responsibilities or lack of provision. Finally, *broader engagement and norms* reflected alignment with collective and institutional measures as well as perceived social expectations (e.g. support for low-carbon policies, perceived attitudes of peers). Though the specific content varied according to the domain, all components were designed to reflect the interaction of individual agency with systemic opportunities and

## Table 7 | Items for assessing capability scores for transportation

| CC Component | *With regard to your transportation, how would you rate the following statements on a scale of strongly agree to strongly disagree?* |
|---|---|
| Attitudes, skills, knowledge | 1. Reducing the environmental impacts associated with my transportation is important to me. |
| | 2. I am willing to reduce how much I travel overall. |
| | 3. I feel comfortable using and navigating public transport (buses and trains). |
| Behaviour and choices | 4. I never use a car for trips under 2 miles. |
| | 5. If I use a car, I share trips most of the time (with family, friends or others). |
| | 6. I do not fly. |
| Structural capacity | 7. There are significant barriers to reducing the environmental impacts associated with my travel (e.g. cost, few public transport options). |
| | 8. There is adequate walking and cycling infrastructure in my local area. |
| | 9. My family/caring/work responsibilities require me to use a car. |
| Broader engagement and norms | 10. It is important for me to be able to travel internationally. |
| | 11. People in the UK are too dependent on cars. |
| | 12. I support policies which encourage more sustainable transport. |

constraints. While the specific content varied by domain, the structure was consistent (6 domains × 4 components × 3 items), enabling cross-sectoral comparison.

Responses were recorded on a seven-point Likert scale. Where agreement indicated greater capability (e.g. 'Reducing the environmental impacts associated with my transportation is important to me'), responses were coded from +3 ('strongly agree') to −3 ('strongly disagree'); for items where agreement indicated lower capability (e.g. 'There are significant barriers to reducing the environmental impacts associated with my travel'), the scale was inverted.

Domain scores were obtained by summing the 12 items (range: −36 to +36). It can be argued that not all items hold equal weight and that some should be given higher weightings in the Score. However, given the subjectivity of weighting different items in determining what constitutes CC, it was decided to hold all questions equal for simplicity and comparability. This approach is consistent with the symmetrical representation of CC in Whitmarsh et al.'s original diagram[14]. Treating the components equally also responds to Hampton and Whitmarsh's critique of the CC literature[16], which has tended to pay greater attention to measures of *agency* compared with *structure* or social norms.

The 72 items across six domains produced an overall capability score ranging from −216 to +216. For reporting, this continuous measure was converted into letter grades (A–F) to distinguish capability scores from numerical CF values (Table 8).

Capability component scores were also calculated by summing the three questions related to that capability in each domain and then averaging across each domain. Thus, CC component scores range from -9 to 9, where 9 is the highest level of capability.

### Carbon footprint

Personal CFs need more rigorous and transparent methodologies. CF calculators make trade-offs between being user-friendly and the quantity and quality of data used to make estimates[24,40]. However, when balancing these trade-offs, Padgett et al.[46] and Birnik[40] found that CF calculators lack consistency, transparency, and data quality. This poses problems for comparing between CFs and making plans of action to change behaviour. Having identified issues such as the excessive reliance on generic datasets and oversimplified assumptions, Birnik[40] proposed thirteen principles for developing CF tools, including the use of multiple GHGs, household size and income adjustments, and up-to-date, region-specific conversion factors. The methodology developed in this study adheres to these principles.

Items used to calculate respondents' CF focused on four consumption-related domains: household energy use, food, transport, and shopping. Responding to Birnik's[40] 13 principles for CF assessments, our methodology

## Table 8 | Capability score by letter grade

| Letter grade | Domain capability score | | Total capability score | |
|---|---|---|---|---|
| | Bottom range | Top range | Bottom range | Top range |
| A | −36 | <−24 | −216 | <−144 |
| B | −24 | <−12 | −144 | <−72 |
| C | −12 | <0 | −72 | <0 |
| D | 0 | <12 | 0 | <72 |
| E | 12 | <24 | 72 | <144 |
| F | 24 | 36 | 144 | 216 |

## Table 9 | Carbon conversion factors for household energy[47]

| Fuel | Unit | kg CO₂e |
|---|---|---|
| Electricity | kWh | 0.20705 |
| Natural gas | kWh | 0.18290 |

emulated—and some domains improved upon—one of the most prominent calculators in the UK, developed by WWF in collaboration with researchers at Leeds University[25,26].

### Household energy

One common limitation of CF calculators is that individuals tend to inaccurately estimate their household energy usage[41]. In response, it is common for calculator developers to use secondary data to model this element of the CF[26]. Our approach employed a predictive model which was trained on data from the Smart Energy Research Lab (SERL) predictive model. SERL comprises energy consumption data from nearly 13,000 UK households, using accurate smart metre readings. In this paper, we employed the novel modelling approach developed and tested by Satre-Meloy and Hampton[31], who applied Least Absolute Shrinkage and Selection Operator regression to a large dataset of socio-demographic variables, building characteristics, and attitudinal and behavioural predictors to produce a simplified model for household energy consumption. To apply this approach, our survey used 24 of the items which were found in the Satre-Meloy and Hampton study to have non-zero coefficients[31]. These were then used to predict annual household energy consumption (gas and electricity) for our survey respondents. Carbon conversion factors published by the UK Government were used to convert annual energy use into CO₂e (Table 9)[47].

**Table 10 | Conversion factors for modes of transportation[47]**

| Mode | CO₂e factor | Metric |
|---|---|---|
| Small petrol (under 0.4 L engine) or diesel car (under 1.7 L engine) | 0.228 | kg CO₂e/mile |
| Medium petrol (1.4–2 L engine) or diesel car (1.7–2 L engine) | 0.278 | kg CO₂e/mile |
| Large petrol (2 L+ engine) or diesel car (2 L+ engine) | 0.383 | kg CO₂e/mile |
| Electric car | 0.076 | kg CO₂e/mile |
| Plug-in hybrid car | 0.175 | kg CO₂e/mile |
| Hybrid car | 0.203 | kg CO₂e/mile |
| Bicycle | 0.000 | kg CO₂e/mile |
| Walking | 0.000 | kg CO₂e/mile |
| Bus | 0.108 | kg CO₂e/mile/ passenger |
| Train | 0.031 | kg CO₂e/mile/ passenger |
| Van | 0.285 | kg CO₂e/mile |
| Taxi | 0.149 | kg CO₂e/km /passenger |
| Motorbike | 0.183 | kg CO₂e/mile |

**Table 11 | Carbon conversion factors for different UK diets (adapted from Scarborough et al.)[50]**

| Diet Group | Women CO₂e (2000 kcal) | Men CO₂e (2500 kcal) | Other gender/Prefer Not to Say CO₂e (2225 kcal) |
|---|---|---|---|
| Vegans | 2.47 | 3.09 | 2.79 |
| Vegetarians | 4.16 | 5.20 | 4.68 |
| Fish-eaters | 4.74 | 5.93 | 5.33 |
| Low meat-eaters | 5.37 | 6.71 | 6.04 |
| Medium meat-eaters | 7.04 | 8.80 | 7.92 |
| High meat-eaters | 10.24 | 12.8 | 11.52 |

$N = 55,504$ adults. Median values are presented, based on GWP100 CO₂e.

## Transportation

Respondents were asked how many hours they spent travelling in different modes (Q25)[32]. To convert hours travelled into mileage, an average speed was estimated based on 2022 road and speed data from the UK Department of Transport (DFT Tables TRA4215; NTS0303; SPE0111). This follows WWF's methodology, which assumes that respondents find it easier to report hours spent travelling rather than distances[25,26]. Carbon conversion factors are expressed in passenger miles/km; however, average speed data were obtained from government sources to produce CO₂e calculations. For WWF, Harris et al.[26] use a figure of 31.1 mph, but do not account for different road types. We used DfT average speeds for minor and major roads, and motorways. Weighting these by total mileage produced an average speed of 39.6mph. Respondents reported hours spent in different types of vehicles (car, van, taxi, and motorbike), which were multiplied by this average speed and their respective carbon emission conversion factors (Table 10). For each car size, petrol and diesel factors were averaged. For electric, plug-in hybrid, and hybrid factors, the factor for an average-sized car was used.

For trains and buses, the WWF ecological footprint calculator assumption was used: 1 h travel time was on average 60 km (37.28 miles) on a train and 30 km (18.64 miles) on a bus. WWF did not include a justification for these average speeds in their methodology[25,26]. A study by the Press Association[48] found a range of average speeds across different train types, including significant regional variations, and analysis of these data suggested the WWF figure was reasonable. Additionally, WWF's bus speed fell in line with what could be extrapolated from urban road average speeds in the UK and the fact that buses are disproportionately used in urban areas (UK DFT Table SPE0111; BUS03). The bus factor used the average local bus conversion factor. The train factor was calculated from the average factor of national rail, light rail/tram, and the London underground. In the 'other' category, some respondents wrote in motorbike, van, or taxi. The factor for an average motorbike was used. Taxis were assumed to be regular taxis. Vans were assumed to be class I (up to 1.305 tonnes) and an average of the diesel and petrol factor was used.

For aviation, respondents were asked about both business and personal travel, and both were included in their CF. While some CF calculators exclude business travel from individuals' CF, CC emphasises individuals' multiple roles in society, and this includes their membership of organisations[16]. The number of hours spent flying was included as a categorical variable, and following the WWF methodology, the midpoint of each category was taken. For those who report flying 100+ hours, they are given 110 h of flying[26]. This approach yields conservative estimates for flying-based emissions. Carbon Independent's factor of 250 kgCO₂e per hour of air travel was used[49]. Carbon Independent's methodology combines two different calculations which arrive at the same factor: (1) calculating emissions from fuel consumption per flight, and (2) calculating emissions from total UK fuel consumption. It also includes radiative forcing.

Lastly, 33 respondents marked 'other' forms of transport. These responses were reviewed individually and most were deemed to be zero emission (e.g. 'wheelchair'). Others were recoded into another category (e.g. 'tube' was recoded as train).

## Food and diet

Survey respondents were asked to classify their diets into six types, using categories from Scarborough et al[50]. That study calculates dietary GHG emissions by UK diet group, aggregated using the GWP100, standardised to 2000 kcal, but broken down by age and gender. We applied the conversion factors calculated by Scarborough and colleagues (Table 11) but adjusted for differences in caloric intake for men and women. The NHS advises that the average man needs 2500 kcal a day and the average woman needs 2000 kcal a day. Those who marked other or prefer not to say for gender ($N = 11$) were given the average calorie intake of 2225 kcal a day.

## Food waste

Respondents were asked to estimate what percentage of household food was wasted, using percentage ranges (Q30)[32]. Due to individuals' tendency to underestimate their food waste[51], the top of the selected range was applied as a multiplier to calculate food waste related emissions. For example, someone who says they waste 1–10% of the food they buy would be assigned a 1.1 multiplier to their diet-based CF. For those who marked that they waste more than 30% of food, a multiplier of 1.4 was assigned. This is consistent with the approach used in the WWF calculator[26].

## Shopping

The last major domain of consumption-based emissions involves the purchase of products and services other than food and energy. Following other CF methodologies, our survey asked respondents about household income, and the proportion of disposable income spent on new products (e.g. clothes, cosmetics, etc.). The UK's ONS publishes data mapping income decile to weekly household expenditure for a range of goods and services (Table 12)[52]. Emissions were calculated using the Classification of Individual Consumption According to Purpose (COICOP) factors which are provided by the UN Statistics Division[53] and widely used to calculate consumption-based emissions[54]. COICOP factors are expressed in units of kgCO₂e/£ (Table 13).

 

**Table 12 | Family spending by household income decile (weekly expenditure, £), adapted from ONS[52]**

| Deciles | 1 | 2 | 3 | 4 | 5 | 6 | 7 | 8 | 9 | 10 |
|---|---|---|---|---|---|---|---|---|---|---|
| Household goods and services | 12.4 | 17.5 | 21.4 | 27.2 | 30.8 | 37.5 | 35.3 | 38.4 | 47.8 | 76.8 |
| Other recreational items and equipment, gardens and pets | 4.7 | 9.9 | 9.7 | 10.4 | 12.5 | 18.7 | 17.8 | 19.9 | 21.5 | 38.3 |
| Audio-visual, photographic and information processing equipment | 0.8 | 2.1 | 3.3 | 1.7 | 4.4 | 6.7 | 7.4 | 5.4 | 6.1 | 9.4 |
| Personal care | 3.5 | 4.7 | 8.3 | 7.9 | 8.8 | 9.9 | 11.9 | 14.1 | 14.3 | 15.8 |
| Newspapers, books and stationery | 2.2 | 3.1 | 4.7 | 3.9 | 4.5 | 4.3 | 5.7 | 6.3 | 5.6 | 8.5 |
| Clothing | 2.7 | 4.9 | 8.4 | 7.5 | 11.1 | 10.4 | 14.2 | 14.3 | 18.6 | 25.4 |
| Footwear | 0.5 | 1 | 2 | 1.8 | 2.5 | 2.1 | 3.2 | 3.8 | 4.8 | 5.8 |
| Purchase of new cars and vans | 0 | 0 | 3.6 | 3.9 | 4.9 | 6.9 | 6.9 | 12.9 | 25.6 | 23 |
| Purchase of second-hand cars or vans | 3.8 | 6.5 | 11.5 | 10 | 18 | 24.7 | 22.3 | 27.2 | 28.1 | 41.5 |
| Purchase of motorcycles and other vehicles | 0 | 0 | 0 | 0 | 0 | 0 | 0 | 5.1 | 3.5 | 2.6 |
| Education fees | 0 | 0 | 3.7 | 2.1 | 1.4 | 5.7 | 2.8 | 8.4 | 3.5 | 51.3 |
| Communication | 11.1 | 12.9 | 15.8 | 17.4 | 21.8 | 22.2 | 23.5 | 25.1 | 27.9 | 31.9 |

**Table 13 | COICOP factors, 2018[53]**

| Average weekly household expenditure £ | kg $CO_2$e per £ |
|---|---|
| Clothing | 0.2584 |
| Footwear | 0.351 |
| Household goods and services | 0.46 |
| Other recreational items and equipment, gardens and pets | 0.971 |
| Audio-visual, photographic and information processing equipment | 0.6 |
| Personal care | 0.231 |
| Newspapers, books and stationery | 0.159 |
| Purchase of new cars and vans | 0.29 |
| Purchase of second-hand cars or vans | 0.29 |
| Purchase of motorcycles and other vehicles | 0.145 |
| Education fees | 0.338 |
| Communication | 0.487 |

**Table 14 | Consumer preference multipliers**

| Spender category | Percentage of disposable income spent on new items | Number of respondents |
|---|---|---|
| Low | ≤10 | 807 |
| Medium | >10 and ≤21 | 548 |
| High | >21 | 666 |

**Table 15 | Chi-squared tests for representativeness of the sample**

| Variable | $X$-squared | Degrees of freedom | $p$ value |
|---|---|---|---|
| Gender | 0.30086 | 1 | 0.583 |
| Region | 0.35377 | 11 | 1.000 |
| Age | 0.40667 | 5 | 0.995 |
| Education | 0.006904 | 2 | 0.997 |
| Race | 0.72546 | 4 | 0.948 |

Income deciles do not fully explain consumption behaviours, however. Respondents' estimates of expenditure on new items were used to account for differences in lifestyle and consumer preferences. Percentages estimated by respondents were divided into terciles (low, medium, or high), and multipliers applied (Table 14). Low spenders were treated as the lowest spenders within their income decile. Medium spenders were treated as spending the average amount for their decile. High spenders were treated as spending the top amount in their income decile. A line of best fit was used to estimate the 1st and 100th centiles. Household shopping CFs were divided by the number of people in the household to produce figures for individuals.

**Variables predicting carbon capability**

To answer RQ2 and RQ3, a variety of statistical analyses were run on the survey. The survey includes a range of items which are used as independent variables in the statistical analysis, including gender, age, income, education, employment, and household ownership. Four questions (see ref. 32, Qs 13–16) were included to measure how knowledgeable respondents are about climate change[32]. For example, we asked respondents 'by which has the UK government committed to reach net-zero carbon emissions?' [(a) 2030, (b) 2035, (c) 2040, (d) 2050]. Multiple choice questions were selected from an initial list of 10, based on feedback from colleagues and through testing with non-experts. A scale of 1–4 is used, based on the number of correct responses. On average, respondents scored 2.19 out of 4.

**Survey dissemination and quality control**

The survey was disseminated in June-July 2024 by survey company Dynata through their online survey platform. The 20-min survey was completed by 2001 respondents, who were compensated for their time. The sample was representative of the UK in terms of gender, age, region, race, and education, using quotas based on the 2021 census (Table 15). Sample representativeness and quality control procedures are outlined in Tables 16 and 17.

**Summary statistics and regressions**

Summary statistics and regression models were run using RStudio (version 2023.12.0) to answer the research questions. All regressions are ordinary least square models, and checks were run to test assumptions (linearity, homoscedasticity, normality, and multicollinearity)[55]. All regressions passed all checks except for homoscedasticity. Heteroscedasticity-consistent standard errors were therefore applied to the models to correct for heteroscedasticity[55].

In regression analyses, gender was coded as being female (1) or not (0). Age was continuous. Income was ordinal by household income decile. Education was ordinal by level of education (e.g. GCSE or lower; A-levels, BTEC, etc.; or college and higher). Employment was coded as a binary of either working (part-time or full-time) or not. Household ownership was coded as a binary of whether someone is an owner or not.

Regression tables include the independent variables in their non-standardised and standardised form to allow ease of interpretation and

**Table 16 | Checks that resulted in elimination from the sample**

| Check | Details | Number of respondents removed |
|---|---|---|
| Attention filter questions | Respondents were asked to mark a certain answer to two attention filter questions integrated into the survey. | 0 [Dynata already removed those who failed] |
| Answering straight across a CC table | If respondents gave the same answer to all twelve questions in a CC domain table, then it was assumed that they were just clicking through without paying attention. | 29 |
| Not including oneself in the household count | Respondents were asked to include themselves when giving the number and ages of people in their household. If they did not include themselves, then they were removed. | 68 |
| No heat system | If respondents selected that they do not have a heat system and then selected that they do have a specific type of heat system, then they were eliminated. | 0 [No respondents failed this check.] |
| Do not adjust the heat | If respondents selected that they do not adjust the heat in their home and then also selected that they do adjust the heat for specific reasons, then they were eliminated. | 0 [No respondents failed this check.] |
| 63 h or more of travel in one mode of transportation in a week[2] | If respondents marked 63 h or more of travel in one mode of transportation in a week, then it was assumed that they did not read the question correctly or were not paying attention. | 19 |
| Hours of transportation summed to over 100 h a week | If respondents' transportation summed to more than 100 h in a week, then it was assumed that they did not read the question correctly or were not paying attention. | 2 |

**Table 17 | Checks that did not result in immediate elimination, but prompted investigation of the respondent's answers**

| Check | Details | Number of respondents removed |
|---|---|---|
| >7 people in a household | If a respondent marked that they had more than 7 members of their household, then their responses were examined in more depth. | 2 (11 respondents were flagged, and 2 were ultimately removed for nonsensical text responses) |
| >3 heating systems | If a respondent marked that they had more than 3 heating systems installed in their home, then their responses were examined in more depth. | 2 (19 respondents were flagged, and 2 were ultimately removed for nonsensical text responses) |

comparison. A 95% confidence interval is used to determine statistical significance. Significant predictors are indicated as: $*p < 0.05$, $**p < 0.01$, $***p < 0.001$.

### Reporting summary
Further information on research design is available in the Nature Research Reporting Summary linked to this article.

### Data availability
The data that support the findings of this study are archived in the University of Bath Research Data Archive (Carbon Capability Survey, Hampton, 2024)[32]. Due to participant consent conditions, access is available to bona fide researchers for related research on request via the repository. Documentation, figure source code, and a full codebook are available alongside the dataset. Further details and access instructions can be found at https://researchdata.bath.ac.uk/1531 or by contacting the Research Data Service at the University of Bath.

### Code availability
The code used to perform the statistical analyses and to generate the results presented in this study is not publicly archived but can be made available by the corresponding author upon reasonable request. Any prospective users should contact the authors directly to discuss access arrangements and intended usage. The results of the analysis can be replicated without the authors' code.

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

## Acknowledgements

This study was supported by the Economic and Social Research Council (ESRC), grant number ES/V015133/1. The funder had no role in study design, data collection and analysis, decision to publish, or preparation of the manuscript.

## Author contributions

A.G., S.H. and L.W. designed the study and developed the survey together. A.G. led the analysis. A.G. and S.H. conducted the literature review and drafted the manuscript. All authors contributed to interpretation of the results, review, and editing of the manuscript. All authors approved the final version of the manuscript.

## Competing interests

The authors declare no competing interests.
