## [Transparent Peer Review file · Communications Sustainability]

Quantifying carbon capability: integrating carbon footprints with social and systemic factors for climate action assessment

Corresponding Author: Ms Alisa Ghura

Version 0:

Decision Letter:

Dear Dr Hampton,

Your manuscript titled "Beyond carbon footprints: a methodology for evaluating carbon capability" has now been seen by 2 reviewers, whose comments are appended below. You will see that they find your work of some potential interest. However, they have raised quite substantial concerns that must be addressed. We would be interested in considering a revised version that fully addresses these serious concerns.

For publication in *Communications Sustainability* to be appropriate, your study must:

- * provide a compelling new tool for assessing individuals' desire and ability to reduce their carbon footprint, with a clear rationale;
- * discuss limitations and uncertainty; and
- * transparently report methods to enable the reproducibility of the findings.

We hope you will find the reviewers' comments useful as you decide how to proceed. Should additional work allow you to address these criticisms, we would be happy to look at a substantially revised manuscript. If you choose to take up this option, please either highlight all changes in the manuscript text file, or provide a list of the changes to the manuscript with your responses to the reviewers.

When resubmitting, please provide a point-by-point response to the reviewers' comments. Please submit your responses as a separate file, distinct from your cover letter where you can add responses to the Editors' comments that you do not want to be made available to the reviewers. Word files are preferred. We recommend that any figures, tables or graphs that are included in the response to reviewers are also included in the main article or Supplementary Information.

If the revision process takes significantly longer than three months, we will be happy to reconsider your paper at a later date, as long as nothing similar has been accepted for publication at *Communications Sustainability* or published elsewhere in the meantime.

Please use the following link to submit your revised manuscript, point-by-point response to the reviewers' comments with a list of your changes to the manuscript text (which should be in a separate document to any cover letter), a tracked-changes version of the manuscript (as a PDF file) and any completed checklist:

Link Redacted

Please do not hesitate to contact us if you have any questions or would like to discuss the required revisions further. Thank you for the opportunity to review your work.

Best regards,

Yann Benetreau, PhD
Deputy Editor, Communications Sustainability
Consulting Editor, Communications Earth & Environment
Nature Portfolio
ORCID: 0000-0002-1897-0887
New York Office

EDITORIAL POLICIES AND FORMAT

If you decide to resubmit your paper, please ensure that your manuscript complies with our editorial policies and complete and upload the checklist below as a Related Manuscript file type with the revised article:

- Behavioural and social science
- Ecological, evolutionary & environmental sciences
- Life sciences

For your information, you can find some guidance regarding format requirements summarized on the following checklist: (<https://www.nature.com/documents/commsj-phys-style-formatting-checklist-article.pdf>) and formatting guide (<https://www.nature.com/documents/commsj-phys-style-formatting-guide-accept.pdf>).

REVIEWER COMMENTS:

Reviewer #1 (Remarks to the Author):

Now that individuals' Carbon Footprints (CFPs) can be calculated, administration, companies, and NGOs are utilizing this capability to promote decarbonization actions, and research is increasing. While this emphasizes individuals' daily consumption choices, the reviewer also shares the concern that it diverts attention from one-off but high-impact actions, or fails to advance systemic transformation.

The Carbon Capability (CC) framework is presented to overcome the conflict between individual focus and system focus, and it includes social influence and political participation. I also find this framework interesting.

However, as to whether this draft itself holds sufficient academic value, I have several doubts.

First, the examination of the Carbon Capability concept is insufficient. Previous studies proposing the Carbon Capability concept are very briefly introduced from Lines 50 to 64, but it is unclear how the Carbon Capability concept is defined in this study, considering the merits and shortcomings of those studies.

From Line 64 onwards, a "Gap" is pointed out, stating that CC lacks quantitative analysis. However, why is quantitative analysis necessary?

Lines 75 to 81 provide supplementary explanations of domains. However, this explanation, linking individual motivation, knowledge, and access to products/services to "reduction of environmental impact," suggests that the CC concept contains a very significant bias and ultimately boils down to methodological individualism.

I also have doubts about the quantitative methodology of this paper. Social impact and political participation are interactions between individuals, individuals and groups, and individuals and institutions, and cannot be evaluated solely based on scoring from individual questionnaires.

In my view, at least the CC concept analyzed in this paper does not resolve the conflict between individual focus and system focus; rather, it seems to be a veneer that adds just a slight "social" flavor to an individual focus

Reviewer #2 (Remarks to the Author):

Beyond carbon footprints: a methodology for evaluating carbon capability
Review report 30.6.2025

Thank you for the opportunity to read this manuscript. It has some merit, but also requires substantial improvement in my

judgment. Please see below my recommendations:

1. The title is misleading in my opinion. The authors claim to bring together the concepts of carbon footprint and carbon capability. It is not the same as going beyond carbon footprints. Or if it is, then anything combined with carbon footprints is going beyond.
2. I think that it is exaggeration to claim that carbon footprint / demand side action research would somehow have avoided the topic of systemic changes. There is a long tradition in carbon lock-in research. In addition, systemic change has not been a topic absent from carbon footprint literature. Moreover, topics like voting, climate literacy, and willingness to act have all been covered in carbon footprint and demand side action research. It feels like the authors are trying to take a shortcut, or have not delved deep enough to the literature.
3. While acknowledging the style guidelines of the journal in question, the authors fail at giving the reader enough information to understand the study without carefully reading the methods section in the end of the manuscript. The authors must tell more about the carbon footprint calculation and the CC assessment to allow the reader to understand, while still providing the details in the methods section in the end. For example, the calculator used and the scope of the assessment, or justification for the categories of CC, cannot be left out when these are first introduced. Even reading the methods part does not reveal if the calculator values are always scopes 1-3 or something else.
4. To me how the authors explain CC is surprising. I have always learnt it to mean the power over one's climate impact, but now the authors explain that it is a combination of that and one's willingness to act.
5. Closely related to #4, if there is no weighing in terms of what is important and what is not, and "individual traits" is included, what anymore makes the score somehow a measure related to capability to act vs. systemic change (if that ultimately is what CC means)? Easy actions are completely different from those that make a difference but require (at least perceived) sacrifice. Similarly, most people nowadays say that they to some extent care about climate, but that says almost nothing about their willingness to take any serious action, even in terms of supporting systemic change.
6. Related to #5, the weak relationship between CC and CF the authors find, previous studies have found a stronger relationship between CF and climate concern, but low levels of engagement in most climate actions be they to reduce one's own footprint or to support systemic change.
7. The discussion section fails to talk about these mentioned previous studies that are too closely related to be ignored. Such broader comparison instead of focusing on showing how the footprint is in line with previous studies on UK footprints would also allow for better positioning of the study at hand with those that partially overlap but lack some components present in this study.

** Visit Nature Portfolio's author and referees' website at www.nature.com/authors for information about policies, services and author benefits**

Communications Sustainability is committed to improving transparency in authorship. As part of our efforts in this direction, we are now requesting that all authors identified as 'corresponding author' create and link their Open Researcher and Contributor Identifier (ORCID) with their account on the Manuscript Tracking System prior to acceptance. ORCID helps the scientific community achieve unambiguous attribution of all scholarly contributions. You can create and link your ORCID from the home page of the Manuscript Tracking System by clicking on 'Modify my Springer Nature account' and following the instructions in the link below. Please also inform all co-authors that they can add their ORCID to their accounts and that they must do so prior to acceptance.

Version 1:

Decision Letter:

Dear Dr Hampton,

Your manuscript titled "Quantifying carbon capability: integrating carbon footprints with social and systemic factors for climate action assessment" has now been seen by 2 reviewers, and we include their comments at the end of this message. They appreciate the revisions and find your work of interest, but some important points are raised. We are interested in the possibility of publishing your study in Communications Sustainability, but would like to consider your responses to these concerns and assess a revised manuscript before we make a final decision on publication.

For publication in Communications Sustainability to be appropriate, your study must report your methods and analyses in sufficient detail to enable the reproducibility and replicability of your findings, as highlighted in Reviewer 1's report.

We therefore invite you to revise and resubmit your manuscript, along with a point-by-point response that takes into account the points raised. Please highlight all changes in the manuscript text file.

Please submit your point-by-point responses as a separate file, distinct from your cover letter where you can add responses to the Editors' comments that you do not want to be made available to the reviewers. Word files are preferred. We recommend that any figures, tables or graphs that are included in the response to reviewers are also included in the main article or Supplementary Information.

Please use the following link to submit your revised manuscript, point-by-point response to the reviewers' comments (which should be in a separate document to any cover letter), a tracked-changes version of the manuscript (as a PDF file) and the completed checklist:

Link Redacted

We hope to receive your revised paper within six weeks; please let us know if you aren't able to submit it within this time so that we can discuss how best to proceed. If we don't hear from you, and the revision process takes significantly longer, we may close your file. In this event, we will still be happy to reconsider your paper at a later date, as long as nothing similar has been accepted for publication at Communications Sustainability or published elsewhere in the meantime.

Please do not hesitate to contact us if you have any questions or would like to discuss these revisions further. We look forward to seeing the revised manuscript and thank you for the opportunity to review your work.

Best regards,

Yann Benetreau, PhD
Deputy Editor, Communications Sustainability
Consulting Editor, Communications Earth & Environment
Nature Portfolio
ORCID: 0000-0002-1897-0887
New York Office

EDITORIAL POLICIES AND FORMATTING

- Behavioural and social science
- Ecological, evolutionary & environmental sciences
- Life sciences

Furthermore, please align your manuscript with our format requirements, which are summarized on the following checklist:

<https://www.nature.com/documents/commsj-phys-style-formatting-checklist-article.pdf>>Communications Sustainability formatting checklist

and also in our style and formatting guide <https://www.nature.com/documents/commsj-phys-style-formatting-guide-accept.pdf>>Communications Sustainability formatting guide .

*** DATA: Communications Sustainability endorses the principles of the Enabling FAIR data project (<http://www.copdess.org/enabling-fair-data-project/>). We ask authors to make the data that support their conclusions available in permanent, publicly accessible data repositories. (Please contact the editor if you are unable to make your data available).

All Communications Sustainability manuscripts must include a section titled "Data Availability" at the end of the Methods section or main text (if no Methods). More information on this policy, is available at <http://www.nature.com/authors/policies/data/data-availability-statements-data-citations.pdf>><http://www.nature.com/authors/policies/data/data-availability-statements-data-citations.pdf>.

If a community resource is unavailable, data can be submitted to generalist repositories such as <https://figshare.com/> or <http://datadryad.org/> Dryad Digital Repository. Please provide a unique identifier for the data (for example a DOI or a permanent URL) in the data availability statement, if possible. If the repository does not provide identifiers, we encourage authors to supply the search terms that will return the data. For data that have been obtained from publicly available sources, please provide a URL and the specific data product name in the data availability statement. Data with a DOI should be further cited in the methods reference section.

REVIEWER COMMENTS:

Reviewer #1 (Remarks to the Author):

After reviewing the previous draft, I noted that the paper was ultimately a study focused on individuals and didn't achieve the goal of bridging individual and systems-level approaches as claimed in the introduction. Another reviewer also seemed to point out that the literature review was insufficient.

In response, the writing team has made a commendable effort to address these concerns and elevate the study's significance. In particular, the analysis method, which draws on Whitmarsh's six domains for Climate Change to analyze "components" of individual traits, choices and behavior, broader engagement with carbon governance and social norms, and structural capacity, should be well-received. The discussion section analyzes the influence of these components across consumption categories like transportation, energy, and food. I believe this provides valuable data for designing appropriate interventions for each sector. The observation that there are both correlations and discrepancies between footprints and capabilities in different sectors is also excellent "food for thought."

Building on these improvements, I would like to propose further enhancements to clarify the paper's strengths.

1. Clarify the Framework of the "Components": How were the components of Individual traits, choices and behavior, broader engagement with carbon governance and social norms, and structural capacity organized in the actual questionnaire and analysis? It's likely that elements classified under "Engagement" or "Structural capacity" would differ significantly across consumption sectors like energy, transportation, and food. The current text doesn't clearly explain this framework. I suggest enriching this explanation using one of the following methods:

a) Methodology Section: Explain how the four components were extracted and categorized in the methodology section.

b) Discussion Section: Elaborate on the descriptions for each consumption sector in the discussion section. For example, in the food sector, is the primary issue a lack of information about the environmental impact of foods, or a lack of low-carbon food options (or their high cost)?

2. Strengthen the Implications and Future Research: The current discussion section contains several valuable insights. I suggest adding a section at the end of the discussion or in the conclusion that outlines the key implications of these findings. This could include policy implications, contributions to the discourse on Carbon Capability, and suggestions for future research to deepen the study's findings.

Reviewer #2 (Remarks to the Author):

I thank the authors for revising the manuscript following my comments. I do still disagree with the weighting issue, but at least the issue is brought up in the discussion about the limitations of the study. I can therefore now recommend publishing.

** Visit Nature Portfolio's author and reviewers' website at www.nature.com/authors for information about policies, services and author benefits**

Communications Sustainability is committed to improving transparency in authorship. As part of our efforts in this direction, we are now requesting that all authors identified as 'corresponding author' create and link their Open Researcher and Contributor Identifier (ORCID) with their account on the Manuscript Tracking System prior to acceptance. ORCID helps the scientific community achieve unambiguous attribution of all scholarly contributions. You can create and link your ORCID from the home page of the Manuscript Tracking System by clicking on 'Modify my Springer Nature account' and following the instructions in the link below. Please also inform all co-authors that they can add their ORCID to their accounts and that they must do so prior to acceptance.

Version 2:

Decision Letter:

Dear Dr Hampton,

Your manuscript titled "Quantifying carbon capability: integrating carbon footprints with social and systemic factors for climate action assessment" has now been seen by our reviewers, whose comments appear below. In light of their advice we are delighted to say that we are happy, in principle, to publish a suitably revised version in Communications Sustainability.

We therefore invite you to revise your paper one last time to address the remaining concerns of our reviewers. At the same time we ask that you edit your manuscript to comply with our format requirements and to maximise the accessibility and therefore the impact of your work.

EDITORIAL REQUESTS:

Please make sure to include a Supplemental Information file if relevant.

*****Please take care to match our formatting and policy requirements. We will check revised manuscript and return manuscripts that do not comply. Such requests will lead to delays. *****

SUBMISSION INFORMATION:

OPEN ACCESS:

Communications Sustainability is a fully open access journal. Articles are made freely accessible on publication. For further information about article processing charges, open access funding, and advice and support from Nature Portfolio, please visit <https://www.nature.com/commssustain/open-access>

Please use the following link to submit the above items:
Link Redacted

Best regards,

Yann Benetreau, PhD
Deputy Editor, Communications Sustainability
Consulting Editor, Communications Earth & Environment
Nature Portfolio
ORCID: 0000-0002-1897-0887
New York Office

REVIEWERS' COMMENTS:

Reviewer #1 (Remarks to the Author):

The authors were very responsive to the second round of revisions, successfully improving the paper to a sufficient level to be recommended for publication. The analysis clearly considers the importance of both individual and systemic factors, which has led to a deeper understanding of the difficulties in the UK's current decarbonization transition.

** Visit Nature Portfolio's author and reviewers' website at www.nature.com/authors for information about policies, services and author benefits**

Response to Reviewers

Reviewer 1

General View of the Paper		
1.1	Now that individuals' Carbon Footprints (CFPs) can be calculated, administration, companies, and NGOs are utilizing this capability to promote decarbonization actions, and research is increasing. While this emphasizes individuals' daily consumption choices, the reviewer also shares the concern that it diverts attention from one-off but high-impact actions, or fails to advance systemic transformation. The Carbon Capability (CC) framework is presented to overcome the conflict between individual focus and system focus, and it includes social influence and political participation. I also find this framework interesting. However, as to whether this draft itself holds sufficient academic value, I have several doubts.	We are grateful to the reviewer for their comments and reflections on the article. The CC framework does indeed help to address the tension between a focus on individual and system change, and below we explain how we have revised the manuscript in response to your detailed comments and suggestions.
Detailed comments		
1.2	First, the examination of the Carbon Capability concept is insufficient. Previous studies proposing the Carbon Capability concept are very briefly introduced from Lines 50 to 64, but it is unclear how the Carbon Capability concept is defined in this study, considering the merits and shortcomings of those studies.	This is a fair critique, and is partly a result of word limit. The conceptual model has been published elsewhere so we wanted to avoid duplication here. Nonetheless, we have added some comment and critique on previous CC studies, as well as a clearer explanation of how we are applying it in this study.
1.3	From Line 64 onwards, a "Gap" is pointed out, stating that CC lacks quantitative analysis. However, why is quantitative analysis necessary?	We agree there was a need for more comprehensive justification in the introduction. We have clarified in the revised manuscript that quantifying carbon capability enables systematic comparison across individuals and groups, supports targeted policy and behavioural interventions, allows for tracking of progress over time, and provides meaningful feedback to individuals and communities. A scoring approach turns an abstract concept into an actionable tool, enhancing both academic analysis and public engagement.

1.4	Lines 75 to 81 provide supplementary explanations of domains. However, this explanation, linking individual motivation, knowledge, and access to products/services to "reduction of environmental impact," suggests that the CC concept contains a very significant bias and ultimately boils down to methodological individualism.	This point echoes some criticism of the carbon capability literature for having tended to focus on the role of individuals, despite its deliberate intention to move beyond methodological individualism. Hampton & Whitmarsh (2024) make this observation and argue that CC research should focus on systems of provision in their own right, rather than through the lens of individuals and the degree to which they engage with structures of governance etc. They also suggest that the actors addressed in the CC framework need not be individuals. We have revised the introduction to explain this critique, but this study does focus on individuals' CC and our data is from a large national survey. However, in designing our methodology we did address this critique by introducing four 'capability components'. Structural capacity is one of these, and results are displayed in Figure 4.
1.5	I also have doubts about the quantitative methodology of this paper. Social impact and political participation are interactions between individuals, individuals and groups, and individuals and institutions, and cannot be evaluated solely based on scoring from individual questionnaires. In my view, at least the CC concept analyzed in this paper does not resolve the conflict between individual focus and system focus; rather, it seems to be a veneer that adds just a slight "social" flavor to an individual focus.	Similar to the last point, this echoes the critique made by Hampton & Whitmarsh (2024) about the entire CC literature to date, which has almost exclusively employed individual questionnaires. We share the opinion that CC research could be enriched by a focus on organisations and institutions rather than individuals, and the author team are involved in parallel research which is seeking to do so. Nonetheless, this paper follows the tradition of empirical CC research using surveys, and add a citation to a key text outlining the value of environmental psychology even for structural change. Our approach adds methodological rigour to existing CC literature by developing a broader set of measures than previous work, across domains and components, and capturing structural elements.

Reviewer 2

General comment

	Thank you for the opportunity to read this manuscript. It has some merit, but also requires substantial improvement in my judgment. Please see below my recommendations	Thank you for your review. We are grateful for your comments and recommendations, which have helped to improve the manuscript.
Detailed comments		
2.1	1. The title is misleading in my opinion. The authors claim to bring together the concepts of carbon footprint and carbon capability. It is not the same as going beyond carbon footprints. Or if it is, then anything combined with carbon footprints is going beyond.	We thank the reviewer for raising the issue of clarity in the paper's title. In response, we have adopted the suggested revision: Quantifying carbon capability: integrating carbon footprints with social and systemic factors for climate action assessment . This title more accurately reflects the multidimensional framework and methodological novelty presented in the manuscript, explicitly signposting the integration of carbon footprint measurements with the wider social and systemic dimensions of carbon capability.
2.2	2. I think that it is exaggeration to claim that carbon footprint / demand side action research would somehow have avoided the topic of systemic changes. There is a long tradition in carbon lock-in research. In addition, systemic change has not been a topic absent from carbon footprint literature. Moreover, topics like voting, climate literacy, and willingness to act have all been covered in carbon footprint and demand side action research. It feels like the authors are trying to take a shortcut, or have not delved deep enough to the literature.	This is a fair critique. It was not our intention to shortcut this literature, but the word count prevents detailed coverage of demand side research which addresses carbon lock in and systemic constraints. In response, we have revised the introduction to better reflect this literature, citing studies linking carbon footprints to literacy and public sphere behaviours, while keeping the text as dense as possible.
2.3	3. While acknowledging the style guidelines of the journal in question, the authors fail at giving the reader enough information to understand the study without carefully reading the methods section in the end of the manuscript. The authors must tell more about the carbon footprint calculation and the CC assessment to allow the reader to understand, while still providing the details in the methods section in the end. For example, the calculator used and the scope of the assessment, or justification for the categories of CC, cannot be left out when these are first introduced. Even reading the	As part of a substantially revised introduction, we have now added a short methodological summary at the end of this section. The GHG Protocol's Scope 1, 2, and 3 categories were developed for organisational reporting and are not directly applicable to household carbon footprint methodologies, which are conventionally structured by consumption domain rather than by scope. The full methodological description, including calculation details, remains in the Methods section as per journal style.

	methods part does not reveal if the calculator values are always scopes 1-3 or something else.	
2.4	4. To me how the authors explain CC is surprising. I have always learnt it to mean the power over one's climate impact, but now the authors explain that it is a combination of that and one's willingness to act.	The role of willingness and motivation in CC has been interpreted in different ways in the literature. The original definition of CC is 'The ability to make informed judgments and to take effective decisions regarding the use and management of carbon, through both individual behavior change and collective action' (Whitmarsh et al., 2009). CC studies since have adapted this definition, and some go further in highlighting the need for motivation alongside ability. This includes Horry et al's (2023) 'Climate Capability' scale, and Wei et al's (2016) 5 stage model of CC. The methodology presented here accommodates motivation and willingness to act to an extent, as part of the component titled 'attitudes, skills, knowledge'. But we did not want to narrow the scope such that pro-environmental values are a prerequisite for CC (as per Wei et al). An individual may still have a high CC score without agreeing that 'reducing the environmental impacts associated with XX is important to me' (the first in the Capability Score Likert-scale items). Most measures in the methodology presented do not focus on motivation.
2.5	5. Closely related to #4, if there is no weighing in terms of what is important and what is not, and "individual traits" is included, what anymore makes the score somehow a measure related to capability to act vs. systemic change (if that ultimately is what CC means)? Easy actions are completely different from those that make a difference but require (at least perceived) sacrifice. Similarly, most people nowadays say that they to some extent care about climate, but that says almost nothing about their willingness to take any serious action, even in terms of supporting systemic change.	This raises an important question relating to the relative weight of different measures of carbon capability, and is something that the author team debated. Some would argue that low-carbon consumption behaviours should be weighted over favourable attitudes or support for policy, while others have highlighted -and even quantified - the importance of public-sphere activities (Sustainable Finance Observatory, 2021). We chose not to weight the capability domains or components, because this would require judgements which could be readily contested. Of course, the choice of measures included also involves judgement, and this is why we chose to incorporate CF analysis, and use four components for Capability Scores to maximise consistency across domains. Taken together, the scores capture individuals' capability to act,

		which is a product of individual traits, social and community factors, and enabling systems. The ease of taking climate action is an interesting perspective. It varies by demographic, and geographical and political context, and indeed highlights the importance of structure. For example, it might be easy to score high for transport in places like the Netherlands where infrastructures encourage active and public transport.
2.6	6. Related to #5, the weak relationship between CC and CF the authors find, previous studies have found a stronger relationship between CF and climate concern, but low levels of engagement in most climate actions be they to reduce one's own footprint or to support systemic change.	The relationship between CC and CF is relatively weak, but statistically significant, as shown in Appendix 4. Standardised coefficients (-0.06 and -0.07) are comparable with age and gender as significant predictors. Noting the reviewer's point about the weak link between concern and action, our measure of CC seeks to go beyond measuring concern, and includes a variety of behaviours across six domains, including willingness to change. While a limitation is that it relies on self-report survey methods, these are ubiquitous amongst CC studies to date.
2.7	7. The discussion section fails to talk about these mentioned previous studies that are too closely related to be ignored. Such broader comparison instead of focusing on showing how the footprint is in line with previous studies on UK footprints would also allow for better positioning of the study at hand with those that partially overlap but lack some components present in this study.	We have revised the discussion section to reflect the new introduction, engaging more directly with studies that have incorporated elements of carbon literacy, political agency, and green investment decisions. While retaining the summary of results and comparison with other footprint studies, we have added a paragraph (penultimately), to position this study in the broader carbon footprint literature, including studies which go beyond consumption. This broader comparative perspective clarifies the distinctive contribution and positioning of our study relative to the existing literature.

Response to Reviewers

Reviewer 1

General View of the Paper	
After reviewing the previous draft, I noted that the paper was ultimately a study focused on individuals and didn't achieve the goal of bridging individual and systems-level approaches as claimed in the introduction. Another reviewer also seemed to point out that the literature review was insufficient. In response, the writing team has made a commendable effort to address these concerns and elevate the study's significance. In particular, the analysis method, which draws on Whitmarsh's six domains for Climate Change to analyze "components" of individual traits, choices and behavior, broader engagement with carbon governance and social norms, and structural capacity, should be well-received. The discussion section analyzes the influence of these components across consumption categories like transportation, energy, and food. I believe this provides valuable data for designing appropriate interventions for each sector. The observation that there are both correlations and discrepancies between footprints and capabilities in different sectors is also excellent "food for thought." Building on these improvements, I would like to propose further enhancements to clarify the paper's strengths.	We thank the reviewer for recognising the revisions we undertook to address concerns about the framing and literature base. We are pleased the reviewer sees value in our sectoral breakdowns and in highlighting both alignments and mismatches between footprints and capability. We agree that these results provide a stronger bridge between i-frame and s-frame approaches and clarify the paper's contribution.
Detailed comments	
1. Clarify the Framework of the "Components": How were the components of Individual traits, choices and behavior, broader engagement with carbon governance and social norms, and structural capacity organized in the actual questionnaire and analysis? It's likely that elements classified under "Engagement" or "Structural capacity" would differ significantly across consumption sectors like energy, transportation, and food. The current text doesn't clearly explain this framework. I suggest enriching this explanation using one of the following methods:	Thank you for this helpful comment. We have revised the manuscript to make the organisation of the four components (individual traits, behaviours/choices, structural capacity, and engagement/norms) more explicit, and to elaborate on how they vary across domains. a) Methodology: In the Capability score section we now clarify that each domain (energy, transport, food, shopping, influence, citizenship) was assessed using 12 items, systematically grouped into the

a) Methodology Section: Explain how the four components were extracted and categorized in the methodology section. b) Discussion Section: Elaborate on the descriptions for each consumption sector in the discussion section. For example, in the food sector, is the primary issue a lack of information about the environmental impact of foods, or a lack of low-carbon food options (or their high cost)?	four components (3 items per component). We explain that while the content of items varied by sector, the component structure was consistent, ensuring comparability across domains. b) Discussion: We revised the discussion by highlighting which component appears to be the dominant barrier in each sector. For example, in transport we identify structural constraints (infrastructure, alternatives); in household energy, high capability is shown for low-cost actions but larger retrofit is limited by financial/structural barriers; in food, knowledge is relatively high but affordability and availability of low-carbon foods remain key issues; and in influence/citizenship, an attitude–behaviour gap is evident, with strong support for government action but low uptake of personal civic behaviours.
2. Strengthen the Implications and Future Research: The current discussion section contains several valuable insights. I suggest adding a section at the end of the discussion or in the conclusion that outlines the key implications of these findings. This could include policy implications, contributions to the discourse on Carbon Capability, and suggestions for future research to deepen the study's findings.	We appreciate the reviewer's suggestion to expand on the implications and future directions of our study. Following this feedback, we have revised the final parts of the Discussion to integrate three new paragraphs that outline (i) the policy implications of our findings, (ii) the conceptual contribution to the carbon capability framework, and (iii) priorities for future research. These replace our previous closing paragraph on methodology/community use, and ensure the paper now ends on clear implications and a forward-looking note.

Reviewer 2

General comment	
I thank the authors for revising the manuscript following my comments. I do still disagree with the weighting issue, but at least the issue is brought up in the discussion about the limitations of the study. I can therefore now recommend publishing.	Thank you for taking the time to review our revised manuscript.